# Study on the Preparation of Estrone Molecularly Imprinted Polymers and Their Application in a Quartz Crystal Microbalance Sensor via a Computer-Assisted Design

**DOI:** 10.3390/ijms23105758

**Published:** 2022-05-20

**Authors:** Jin Liu, Xuhong Cai, Junbo Liu, Dadong Liang, Kaiyin Chen, Shanshan Tang, Bao Xu

**Affiliations:** 1College of Resources and Environment, Key Laboratory of Straw Comprehensive Utilization and Black Soil Conservation, Ministry of Education, Jilin Agricultural University, Changchun 130118, China; liu777456@163.com (J.L.); 13943631799@163.com (X.C.); liangdadong@aliyun.com (D.L.); 2Jilin Guangxin Engineering Technology Consulting Co., Ltd., Changchun 130022, China; kaiyin001@126.com; 3Institute of Mathematica, Jilin Normal University, Siping 136000, China; xubxo97@163.com

**Keywords:** estrone, molecularly imprinted polymers, selectivity, computational design, quartz crystal microbalance

## Abstract

Computer simulations are widely used for the selection of conditions for the synthesis of molecularly imprinted polymers and can rapidly reduce the experimental cycle time and save labor and materials. In this paper, estrone molecularly imprinted polymers (E1-MIPs) are designed at the M062X/6-311+G(*d*,*p*) level with itaconic acid (IA) as the functional monomer. The imprinted molar ratio between E1 and IA was optimized, cross-linkers and solvents were screened, and the nature of interactions between E1 and IA was explored. The simulated results showed that pentaerythritol triacrylate was the best cross-linker. Meanwhile, when the imprinted molar ratio between E1 and IA was 1:4, the E1–IA complex had the largest amount of hydrogen bonds, the lowest binding energy, and the strongest stability. Using the simulation results as guidance, the E1-MIPs were prepared to modify the electrons of a quartz crystal microbalance (QCM) sensor. The experimental studies showed that the E1-MIPs-QCM sensor had the highest adsorption capacity to E1 in comparison with their analogues, and the lowest detection value of the sensor was 16.00 μg/L. The computer simulations and experimental studies could provide guidance for synthesize novel E1-MIPs materials. It also could provide important references and directions for the application of E1-MIPs.

## 1. Introduction

Estrone (E1, Figure 1) is an important environmental estrogen that interferes with the reproductive system of aquatic organisms even at extremely low concentrations in environmental water, causing glandular atrophy, functional feminization, and altered immune function in males [1]. Not only that, E1 can also harm the human reproductive function through the food chain, inhibit the quantity and quality of sperm, and hinder the growth of egg cells. At the same time, it also has a non-negligible impact on the increase in the incidence of female breast cancer and uterine cancer [2].

Currently, the main detection methods for E1 are liquid chromatography [3] and liquid chromatography-mass spectrometry [4], but in practice, the above detection methods have disadvantages, such as long detection cycles, being relatively expensive, and requirement of pretreatment. However, the E1 pollutant in environmental water is coexists easily with various pollutants and its concentration is very low. At present, the conventional solid-phase extraction adsorbents (silica gel, C_8_, diatomite, and C_18_) lack specific identification to the determinand. Therefore, many components of the analytical sample matrix will be simultaneously extracted. In addition, extraction columns can usually be used only once and their reproducibilities are poor. Especially, for all the detection methods, the determinands should be collected at the monitoring site and then analyzed in the laboratory. Thus, for the traditional analysis methods, the analysis cycle is long, the cost is high, and the steps are tedious. Molecular imprinting technology (MIT) refers to the process of preparing molecularly imprinted polymers (MIPs) that could match the spatial structure of the template molecule via the polymerization reaction by using the specific target substance (template molecule), suitable functional monomer, cross-linker, and initiator. Compared with the conventional adsorbents used in the pretreatment of the above assays, MIPs have high selectivity, good thermal stability, and high mechanical strength, and also have the advantages of scale preparation, easy operation, and low synthesis cost [5,6]. Theophylline MIPs were first prepared using a non-covalent method by Mosbach et al. [7] in 1993, which greatly promoted the development of MIT. Now, MIPs are widely used in sensors [8,9], chromatographic separation [10,11], and solid-phase extraction [12,13]. With the rapid development of quantum chemistry, more and more researchers are rapidly screening functional monomers, cross-linkers, solvents, and optimal imprinted ratios by computer simulation of MIPs systems [14,15,16]. For example, the MIPs of hydroxyzine and cetirizine were designed by a computer-aided design using the B3LYP method and the selectivity coefficients of the potential sensors was predicted based on the simulation results of MIPs [17]. In addition, theoretical calculations can reveal the nature of molecular imprinting and the relationship between the structure and properties of complex formed from a template molecule and functional monomer. Saied M et al. [18] simulated the geometry of diacetyl platinum (II) complexes at the B3PW91 level and analyzed the chemical bonding nature of complexes using atoms in molecules (AIM) theory.

In previous studies related to the preparation of E1-MIPs, methacrylic acid (MAA) and acrylamide (AM) were mainly used as functional monomers, ethylene dimethacrylate (EDMA) as cross-linker, and methanol as solvent [19,20]. There are less systematic theoretical studies related to the other novel functional monomers, crosslinkers, and solvents of E1 molecular imprinting systems. Especially, there are nearly no quantitative or semi-quantitative studies on the mechanism and nature of the imprinting action for E1 molecular imprinting systems. Therefore, theoretical and experimental studies of E1 molecular imprinting systems need to be improved.

Recently, MIPs have been used to assist quartz crystal microbalance (QCM) to determine many determinands. In order to obtain a better selectivity, the QCM sensor used MIPs as the recognition component. The MIPs-QCM sensor can be constructed via the MIP-modified electrode surface of the QCM sensor. When the MIPs are used as the sensitive identification component of the MIPs-QCM sensor, it should exhibit a high selectivity and accuracy, due to its specific adsorption to the target molecule.

In this paper, we investigated the natural bonding orbital (NBO) charges and molecular electrostatic potential (MEP) of E1 and itaconic acid (IA) with the help of DFT to analyze the bonding interaction active sites and screen the optimal interaction ratio, cross-linker, and solvent of the E1–IA imprinting system. The nature of interaction between E1 and IA is explored by the AIM theory. The E1-MIPs were prepared according to the guide of simulation results. The E1-MIP-modified QCM sensor is constructed to detect E1.

## 2. Results and Discussion

### 2.1. Selection of Calculation Methods

Table 1 lists the structural parameters of E1 simulated by the six methods (B3LYP, CAM-B3LYP, M062X, PBE0, and ωB97XD) and two basis sets (6-311+G(*d*,*p*) and 6-31G(*d*,*p*)) as well as the experimental data [21]. The data present in Table 1 show that the structure parameters of E1 calculated by the six methods with two different basis sets were very close, where the deviations of bond lengths were in the range of 0.0033–0.0034 nm by six methods with the 6-311+G(*d*,*p*) basis set. The standard deviations values of the bond angles were in the range 2.3588°–2.4944°. Compared with the other five methods, the deviation of structural parameters calculated by the M062X method was the smallest, which is closer to the experimental crystal data values. It can also be seen from Table 1 that the deviation values of the bond lengths and bond angles of the two basis sets (6-311+G(*d*,*p*) and 6-31G(*d*,*p*)) under the same method were less than 0.0005 nm and 0.14°, respectively, indicating that the effect of the basis set on the bond lengths and bond angles of the E1 stable structure was small. However, the standard deviation values of bond length and bond angle (0.0034 nm and 2.3643°) for 6-311+G(*d*,*p*) were smaller compared to those for the 6-31G(*d*,*p*) basis set (0.0033 nm and 2.3588°) at the M062X level. Therefore, the M062X/6-311+G(*d*,*p*) method was chosen to optimize the geometric configuration of E1, IA, and E1–IA complexes in this study.

### 2.2. Selection of Calculation Methods

According to the calculation results of NBO and MEP, the active sites of E1 and IA were determined. As can be seen from Figure 2, the more negative charges were mainly distributed on the O atoms in the carbonyl and phenolic hydroxyl group, and the negative charge values were −0.558 (O19) and −0.687 (O20) for E1, respectively. The more positive charges were mainly located on the H atoms in the cyclopentanone, methyl, benzene rings, and hydroxyl groups, and the positive charge values were 0.237 (H21), 0.221 (H32), 0.225 (H33), and 0.471 (H42), respectively. Among them, the charges of O20 and H42 on the hydroxyl group were larger in absolute value. Thus, they were the main dominant active sites of E1. The main active sites of IA were O6 (−0.610), O13 (−0.612), H8 (0.495), and H15(0.492), which were the O atom in the carbonyls and the H atom in the hydroxyl group.

As seen from the MEP scale in Figure 2, the electrostatic potential in different regions of the E1 and IA molecules is shown by different colors. The blue region means the atoms have more positive charges, and the red region means the atoms have more negative charges. From Figure 2a, one can find that the positive charges of E1 were mostly centered on H21, H32, H33, and H42. The H42 atom on the hydroxyl group had more positive charges and easily obtained electrons from nucleophiles. The negative charges were centered on O19 and O20, which easily lost electrons. Similarly, H8 and H15 on the hydroxyl group in the IA molecule (Figure 2b) could be used as electron acceptors, and O6 and O13 of the carboxyl group could be used as electron donors. The results of the MEP analysis were consistent with the results of the NBO charge analysis.

### 2.3. Optimization of the Molar Ratio between E1 and IA

The specific affinity and selectivity of MIPs for substrates relies mainly on the highly ordered steric structure formed by functional monomers in MIPs. This highly ordered steric structure is closely related to the interaction molar ratio between template molecules and functional monomers. The larger the molar ratio between template molecules and functional monomers is, the more ordered the steric structure of the polymer is, and the more stable the conformation is. Thus, the synthesis of MIPs with higher affinity, selectivity, and stability is promising. For this reason, the stable conformations of the complexes formed between E1 and IA with different molar ratios (1:1 to 1:4) were calculated in this study. From Figure 3 and Table 2, one can find that the amount of hydrogen bonds were 2, 5, 7, and 8, respectively, when the imprinted molar ratios between E1 and IA were 1:1, 1:2, 1:3, and 1:4. The hydrogen bond lengths formed ranged from 0.1706 to 0.2563 nm. They were in the range of hydrogen bond [22]. It was obvious that, as the molar ratio between E1 and IA increased, the amount of hydrogen bonds also increased gradually. Additionally, when the interaction molar ratio of E1–IA reached 1:5, the stable complex could not be formed. Therefore, the optimal molar ratio between E1 and IA was 1:4.

According to Table 2, the binding energies of E1–IA complexes with different molar ratios (1:1, 1:2, 1:3, and 1:4) were −31.73 kJ/mol, −76.24 kJ/mol, −114.52 kJ/mol, and −141.55 kJ/mol, respectively. With the increase in the molar ratio between E1 and IA, the binding energy of the E1–IA complex decreased, which indicates that the stability of the complex formed by the template molecule and functional monomer increased. The binding energy was the lowest when the molar ratio between E1 and IA was 1:4, which was expected to form more stable complexes. The results of the binding energy calculations were consistent with the conclusion of the complex stability predicted by the amount of hydrogen bonds in the complex model.

### 2.4. Screening of Cross-Linker Agent

The cross-linking degree of MIPs is influenced not only by the amount of cross-linker, but also by its type. The choice of cross-linker also plays a crucial role in the morphology and mechanical stability of MIPs. To achieve a better imprinting effect, the interaction between the selected cross-linker and IA functional monomer should be greater than that between the cross-linker and E1 when preparing MIPs [23], so that the polymer can have pores with a defined spatial configuration. Additionally, at the same time, the functional residues derived from IA could be ordered in the highly cross-linked E1-MIP stereospore pores. Figure 4 shows the binding energy between three CAs (ethylene dimethacrylate, EDMA, pentaerythritol triacrylate PETA, and trimethylolpropane trimethacrylate, TRIM) and E1 (or IA) at a reaction ratio of 1:1. As shown in Figure 4, the binding energies between the three CAs and E1 were all higher than those of the three CAs and IA, which indicates that all three CAs were suitable for E1-MIPs. Among them, PETA had the highest binding energy with E1 (−12.16 kJ/mol) and the lowest binding energy with IA (−44.62 kJ/mol), which indicates that PETA had the weakest interaction with E1 and the strongest interaction with IA. Thus, PETA was the most suitable CA for the preparation of E1-MIPs.

### 2.5. Selection of Solvents

In the synthesis of MIPs, the solvent not only acts as a dispersing medium to disperse all components of the polymerization reaction into a homogeneous phase, but also determines the strength of the non-covalent interaction between the template molecule and functional monomer, the polymer morphology, and the polymer-specific adsorption and selectivity properties. Table 3 shows the solvation energy (Δ*E*_3_) and the C11-H33---O48 hydrogen bond length of the E1–IA stable complex (1:1) in five solvents—water (H_2_O), acetonitrile (ACN), methanol (MT), tetrahydrofuran (THF), and methylbenzene (TL)—which were calculated according to Equation (3). The shorter hydrogen bond length and the higher solvation energy meant that the solvent had less influence on the interaction between the E1 and IA. From the data in Table 3, it could be seen that the Δ*E*_3_ values of the E1–IA stable complex in different solvents was: TL > THF > MT > ACN > H_2_O. The larger Δ*E*_3_ value led to a lower polarity of solvent and less interference on the interaction between the imprinted molecule and functional monomer, which could significantly improve the specific adsorption and selectivity of MIPs. Among the above five solvents, TL had the largest Δ*E*_3_ value (−28.33 kJ/mol) and the C11-H33---O48 had the shortest bond length (0.2412 nm) in the E1–IA stable complex. This indicates that TL had the least influence on the interaction between E1 and IA. Thus, it was judged that TL was the preferred solvent for the synthesis of E1-MIPs, followed by THF, MT, ACN, and H_2_O.

In the preparation of MIPs, the solvent must be able to dissolve the various reagents required in the polymerization reaction. The solubility and adsorption properties of different reaction components (E1, IA, PETA, and AIBN) in different solvents (H_2_O, ACN, MT, THF, and TL) were investigated. The initial concentration of E1 in the experiment was 80 mg/L. As shown in Table 4, all the reaction components were only soluble in ACN and MT, but slightly soluble in other solvents. The theoretical results suggested that TL was the best solvent; however, the reactant components were not soluble in it. It could also be seen from Table 4 that the amount of adsorption was greater when MT was used as a solvent. Therefore, MT was chosen as the solvent for the preparation of E1-MIPs in this experimental study. Meanwhile, the solvation energy of MT was greater than that of ACN, further verifying the consistency of theory and experiment.

### 2.6. The Nature of Imprinting Interaction

According to the AIM electron density topological analysis theory proposed by Bader [24], the nature of the interaction between E1 and IA was analyzed. From the molecular diagram of the E1–IA stable complex (1:1) (Figure 5a), one can find that there was a bond critical point (BCP) between O20 on the hydroxyl group in E1 and H50 on the hydroxyl group in IA, and the BCP connected O20 and H50 through two bond paths. Similarly, there was a BCP between H33 on the benzene ring in E1 and O48 on the carbonyl group in IA, and the BCP connects H33 and O48 through two bond paths. It indicates that there was a bonding interaction between E1 and IA. From the Laplacian values of the electronic density in Figure 5b, it could be seen that, in the E1–IA stable complex, the O20 atom of E1 and its point of electron concentration as well as the H50 atom of IA and its point of electron deconcentration were in a line. Similarly, the O48 atom of IA and its point of electron concentration as well as the H33 atom of E1 and its point of electron deconcentration were in a line. The calculated results satisfy the geometrical structural features of hydrogen bond formation, which provide evidence for the formation of hydrogen bonds between IA and E1.

In order to quantitatively describe the influence of hydrogen bonding characteristics and hydrogen bond formation on the E1–IA stable complex (1:1), their charge density (*ρ*(r)_bcp_), the Laplacian value (▽^2^*ρ*(r)_bcp_), and the energy density of electrons (EH) of hydrogen bonding at BCP were calculated, respectively. The results are listed in Table 5.

According to reference [24], the *ρ*(r)_bcp_ and ▽^2^*ρ*(r)_bcp_ values at BCP could reflect the nature of the chemical bond. The larger the value of *ρ*(r)bcp at BCP, the stronger the chemical bond formed. When the ▽^2^*ρ*(r)_bcp_ value was larger than 0, the chemical bond was a hydrogen bond. In addition, the energy density (*V*(r)) of electrons also predicts the strength and nature of hydrogen bonds with the energy relationship *E*_H_ = 0.5 *V*(r), and the hydrogen bond energy should be no less than −42 kJ/mol. As can be seen from Table 5, the ▽^2^*ρ*(r)_bcp_ values at BCP between H50---O20 and H33---O48 were 0.1229 and 0.0353 a.u., respectively. It satisfied the requirement for forming a hydrogen bond, which proposed by Rozas et al. [25]. The *ρ*(r) values at BCP were 0.0304 and 0.0105 a.u., respectively. The *ρ*(r)_bcp_ values between H50 and O20 were larger than those between H33 and O48, indicating that the strength of hydrogen bonding between the former was larger than that of the latter. It was consistent with the calculated results of hydrogen bond length. In addition, the *E*_H_ values (−35.58 and −8.27 kJ/mol) of interaction sites were greater than −42 kJ/mol. All the above calculations indicate that E1 interacts with the IA through hydrogen bonding.

### 2.7. Response Characteristics of QCM Sensors

E1-MIPs (NIPs) were synthesized at 333 K, when PETA was as cross-linker, MT was as solvent, and the imprinting ratio between E1 and IA was 1:4. The microspheres of E1-MIPs and NIPs were homogeneous microspheres and uniform distribution (Figure 6) according to the results of scanning electron microscopy (SEM). The distributions and sizes of the microspheres were calculated by the Nano Measurer 1.2 program [26]. The particle sizes ranges of E1-MIPs and NIPs were 305–600 nm (average particle size 459 nm) and 260–530 nm (average particle size 368 nm), respectively. The response values of E1-MIPs and NIP-modified QCM sensors were examined when the E1 standard solutions concentrations are 40, 60, 80, 100, 150, 200, 250, and 300 μg/L, respectively, with the pH value of the PB background solution being 7, the addition of E1-MIPs being 30 mg, the coating amount of PVC being 20 μL, and the response time being 16 min. The corresponding E1 adsorption amounts were calculated and the results are shown in Figure 6. The response frequency of sensor was enhanced as the concentration of E1 increased, corresponding to an increase in the amount of E1 adsorbed. The frequency variation of the E1-MIP-modified QCM sensor was much higher than that of the NIP-modified sensor. Additionally, the difference in adsorption effect increased with increasing the concentration of E1. The regression equation for the response frequency of the E1 molecularly imprinted QCM sensor over the detection concentration range was *Y* = −2.6255*X*−12.685 (*R*^2^ = 0.9983). The low of detection (LOD) could be calculated by the equation LOD = 3.3 *δ*/*S*. *δ* denotes the residual standard deviation of the regression line and S denotes the slope of the regression line. Thus, the LOD value is 16.00 μg/L for the present E1 molecularly imprinted QCM sensor. In comparison with the QCM sensor for the rapid determination of tyramine content in food [27], the E1-MIP-modified QCM sensor achieved the lower value of LOD.

### 2.8. Selectivity of QCM Sensors

To investigate the selectivity of the E1 molecularly imprinted QCM sensor for E1, the response frequencies of the QCM sensor to E1 and its structural analogs (E3 and DES, Figure 7) were measured, respectively. As can be seen from Figure 8, the QCM sensor has a certain response to E1, E3, and DES, and the response frequency was in the order of E1 > E3 > DES. Among them, the highest frequency of sensor response to E1 was due to the presence of functional groups complementary to the E1 functional group in E1-MIPs, thus the sensor response to E1 has the highest value. In addition, both E3 and E1 contained the same benzene ring and cyclopentanone ring with the same reactive groups and similar structures, which were relatively better matched with the imprinted holes of E1-MIPs, so the response frequency of E1 molecularly imprinted QCM sensors to E3 was greater than that to DES.

Moreover, with the increase in reaction time, the response frequency also increases gradually. When the reactive time reached 16 min, the response frequency tends to be stable. At this time, the QCM sensor has the maximum response frequency to E1 (−287.68 Hz). The α values of E3 and DES were 1.65 and 2.47, respectively. It shows that the sensor had high selectivity to E1. The above results indicated that the E1 molecularly imprinted QCM sensor has a good selective recognition ability for E1, and also demonstrated the specific adsorption of MIPs.

## 3. Materials and Methods

### 3.1. Materials and Instruments

E1, Estriol (E3), diethylstilbestrol (DES), and pentaerythritol triacrylate (PETA) were purchased from Shanghai Aladdin Reagent (Shanghai, China). The purities of E1, E3, and DES are over 98%. IA was purchased from Sinopharm Chemical Reagent (Shanghai, China). Azobisisobutyronitrile (AIBN) was purchased from Tianjin Guangfu Technology Development (Tianjin, China). Tetrahydrofuran, methanol, and acetic acid were obtained from Beijing Chemical Works (Beijing, China). Polyvinyl Chloride (PVC, 1000 polymerization) was purchased from China Petroleum & Chemical Corporation, Qilu Branch (Zibo, China).

Tu-1950 Dual light UV-visible Spectrophotometer was purchased from General Instrument (Beijing, China). CHI400C type quartz crystal microbalance sensor was obtained from Chenhua Instruments (Shanghai, China). Scanning electron microscopy (SEM) images of the E1-MIPs and NIPs microspheres were carried out on a JSM-5600 Scanning Electron Microscope obtained from Japan JEOL Electronics Corporation (Tokyo, Japan).

### 3.2. Calculation Method

In order to select a proper calculation method, the B3LYP, CAM-B3LYP, LC-WPBE, ωB97XD, M062X, and PBE0 methods with the 6-311+G(*d*,*p*) and 6-31G(*d*,*p*) basis sets were used to optimize the geometry of E1 (Figure 1). All the calculations were performed with the help of Gaussian 09 program Revision version A.02 software [28].

The bonding sites of E1 and IA were analyzed based on the NBO charge and MEP. When the binding energies were calculated, the iterative addition error of the basis functions was corrected using the counterpoise procedure (CP) method proposed by Boys and Bornardi [29]. The single point energy was calculated first, and then the interaction binding energy (Δ*E*_1_) between E1 and IA was calculated using Equation (1).
Δ*E*_1_ = *E*c − *E*_E1_ − ∑*E*_IA_(1)

*E*_C_ is the total energy of the E1–IA stable complex after correction by the CP method. *E*_E1_ represents the energy of template, and *E*_IA_ is the sum of energy for monomers.

The binding energy Δ*E*_2_ between E1 (or IA) and cross-linker agent (CA) was calculated using Equation (2):Δ*E*_2_ = *E*c − *E*_E1 or IA_ − *E*_CA_(2)

*E*_C_ is the total energy of the complex (1:1) formed by E1 (or IA) and cross-linker after correction by CP method. *E*_E1 or IA_ is the energy of E1 (or IA). *E*_CL_ is the energy of the cross-linker.

Finally, the solvent effect on the imprinting system was evaluated using the self-consistent reaction field (SCRF) approach with the polarizable continuum model (PCM) at 333 K. The stable structure of the complex formed when the molar ratio of E1 to IA was 1:1 was placed in the solvent model and the solvation energy (Δ*E*_3_) was calculated using Equation (3):Δ*E*_3_ = *E*_S_ − *E*_v_(3)

*E*_S_ is the interaction energy between E1 and IA in a solvent environment. *E*_V_ is the interaction energy between E1 and IA under vacuum conditions.

### 3.3. Study on the Nature of Imprinted Polymerization

To reveal the nature of the imprinting action, the charge density and Laplacian values at the bond critical point (BCP) of the imprinted system were performed based on the atoms in molecules (AIM) theory by using the AIM2000 program.

### 3.4. Preparation of E1-MIPs and NIPs

A total of 0.5 mmol (135 mg) E1 and 2 mmol IA were dissolved in 50 mL methanol solvent under ultrasound. A total of 0.1 mol PETA and 0.15 g AIBN were added in the above solution under ultrasound until the powders were completely dissolved. The nitrogen was aerated to mixed solution for 5 min. Then, the mixed solution was sealed in a water bath at 323 K for 24 h. The resulting polymer was extracted repeatedly with methanol/acetic acid solution (8/2, *V*/*V*) using a soxhlet extractor at 343 K to completely remove E1 of the polymer. The residual acetic acid solution of the polymer was removed by extraction with a methanol solution for 24 h at 343 K. The obtained E1-MIPs was put in the vacuum drying oven at 333 K drying to constant weight. The NIPs were prepared as above except that E1 was not added.

### 3.5. Characterization of E1-MIPs and NIPs by Scanning Electron Microscopy

The E1-MIPs (NIPs) was dispersed in methanol, and the mixed solution was put in ultrasonic for 40 min to make the polymer evenly dispersed in methanol. Then, the mixed solution was dropwise dripped onto the silicon wafer and let the silicon wafer dry naturally. After gold spray, the morphology and particle size of E1-MIPs (NIPs) were observed by SEM.

### 3.6. Electrode Pretreatment

The QCM electrode surface was washed with piranha solution (30% H_2_O_2_: 98% H_2_SO_4_ = 1:3, *V*/*V*) for 5 min. Then, the QCM electrode was washed with distilled water. The above process was repeated 3–5 times. The QCM electrode was dried with nitrogen and stored in a drying oven for backup.

### 3.7. Construction of Molecularly Imprinted QCM Electrodes

A total of 30 mg E1-MIPs, 20 mL tetrahydrofuran, and 20 mg PVC were placed in a 50 mL conical flask under ultrasound for 1 h. The above solution was instilled on the surface of the QCM electrode and let stand for 1 h at room temperature to obtain the E1-MIP-modified molecularly imprinted QCM electrode. The NIP-modified QCM electrode was prepared in the same way as described above.

### 3.8. Response Characteristics of the QCM Sensors

The basic principle of the QCM sensor is that, when a modified film on the surface of the electrode adsorbs the substrate leading to a change in its weight, the frequency of the quartz crystal substrate changes accordingly. Then, it is transmitted to the computer. Using the CHI400 software to analyze and obtain the frequency change signal. The QCM electrode has a reference frequency of 8 MHz and a diameter of 8 mm.

The modified QCM electrode was fixed on the detection cell. A total of 2 mL phosphate buffer (PB, pH = 7) was added to the detection cell as the background solution. The initial frequency (*f*_0_) of the sensor was recorded when the frequency stabilized. Then, 10 μL E1 solutions with different concentrations (40, 60, 80, 100, 150, 200, 250, and 300 μg/L) was injected into the background solution, respectively. The response frequency (*f*_i_) was recorded after the frequency stabilized. The frequency change (Δ*f*) of the response was calculated according to Equation (4).
Δ*f* = *f*_1_ − *f*_0_(4)

### 3.9. The Selectivity of QCM Sensor to E1

The response frequencies of E1 and its structural analogs (E3 and DES, Figure 2) were measured by the QCM sensor to investigate the selectivity of the sensor constructed by E1-MIP-modified QCM electrode. The initial concentrations of the E1, E3, and DES standard solutions were 100 μg/L. In addition, the selectivity coefficient (*α*) was calculated on the basis of the formula:*α* = Δ*F*_E1_/Δ*F*_A_(5)

Δ*F*_E1_ (Hz) is the response frequency of sensor to E1 and Δ*F*_A_ (Hz) means the response frequency of sensor to analogues (E3 and DES) of E1.

## 4. Conclusions

The M062X/6-311+G(*d*,*p*) method of DFT was used to simulate the bonding sites and spatial configuration of the complexes formed by E1 as the template molecule and IA as the functional monomer, and optimize the imprinting molar ratio, cross-linker agent, and solvent. The nature of the bonding interaction was further investigated by the electron density topology analysis of the complexes formed by E1 and IA. The theoretical results indicated that E1 interacted with IA through hydrogen bonding, and E1-MIPs with higher specific adsorption could be prepared with PETA as the CA with the molar ratio of 1:4. The experimental results show that the E1 molecularly imprinted QCM sensor has a good selective recognition ability, which was consistent with the theoretical study. The LOD of sensor was 16.00 μg/L. Selectivity studies show that E1-MIPs-QCM sensor has a higher selectivity to E1 than those of E3 and DES. In this study, computer simulation was used to optimize the synthesis of E1-MIPs, which not only shortened the reaction time and reduced energy consumption, but also improved the synthesis and development efficiency of MIPs. In addition, this study also provides theoretical and experimental reference for the selective separation and detection of E1 in samples using the E1-MIPs-QCM sensor.

## Figures and Tables

**Figure 1 ijms-23-05758-f001:**
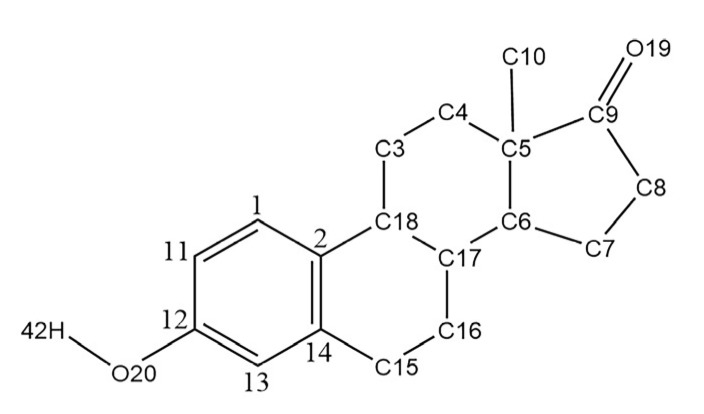
The molecular structure of E1.

**Figure 2 ijms-23-05758-f002:**
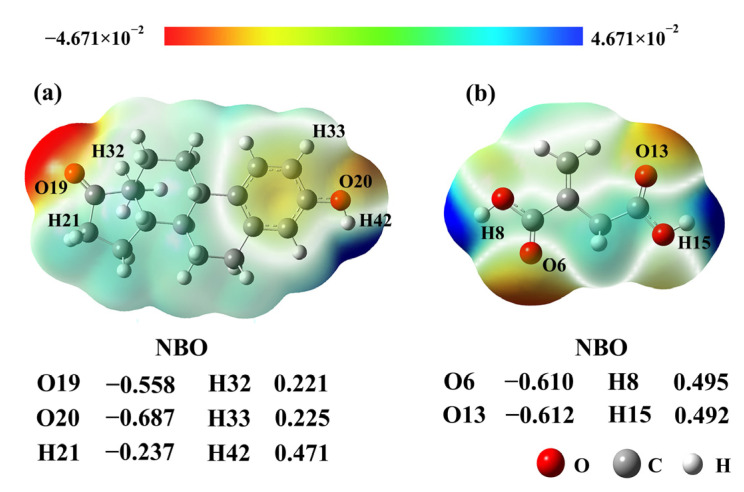
MEP distributions and NBO charges of E1 (**a**) and IA (**b**).

**Figure 3 ijms-23-05758-f003:**
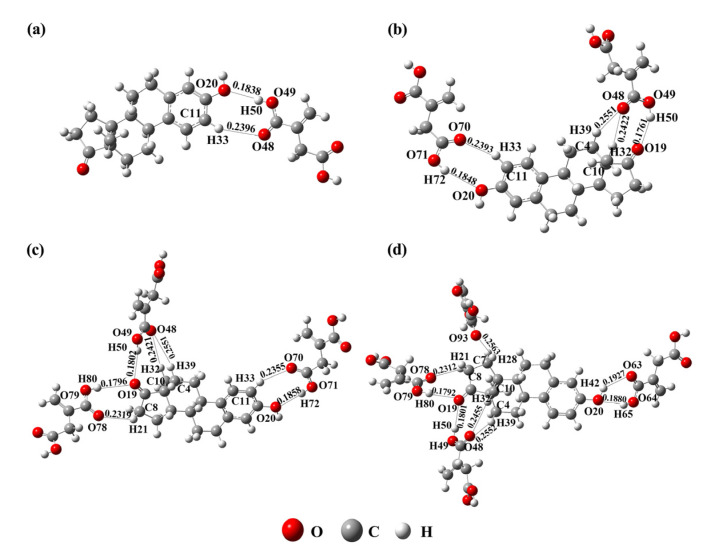
Complexes formed between E1 and IA with different imprinted ratios: 1:1 (**a**), 1:2 (**b**), 1:3 (**c**), and 1:4 (**d**).

**Figure 4 ijms-23-05758-f004:**
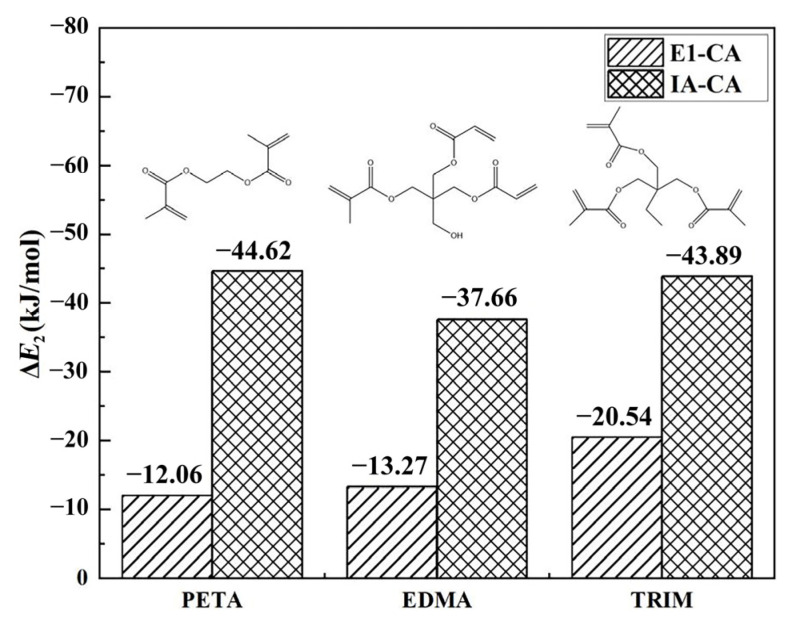
Δ*E*_2_ values between E1 and CA as well as IA and CA.

**Figure 5 ijms-23-05758-f005:**
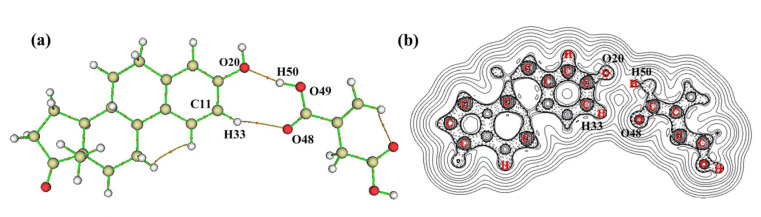
Molecular graphs (**a**) and contours of the Laplacian values of the electronic density (**b**) of the complex formed between E1 and IA.

**Figure 6 ijms-23-05758-f006:**
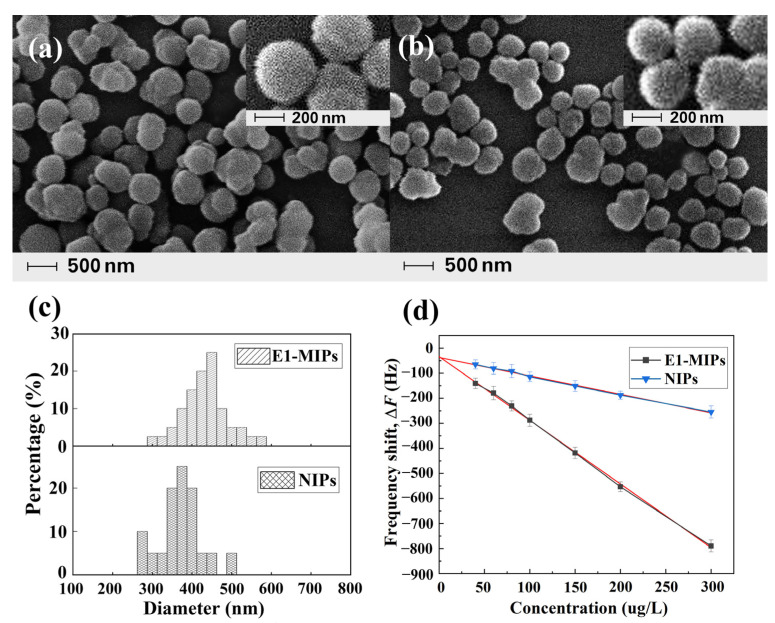
SEM of E1-MIPs (**a**) and NIPs (**b**), particle size distributions of E1-MIPs and NIPs (**c**), and response frequency of the sensor for E1 at different concentrations (**d**).

**Figure 7 ijms-23-05758-f007:**
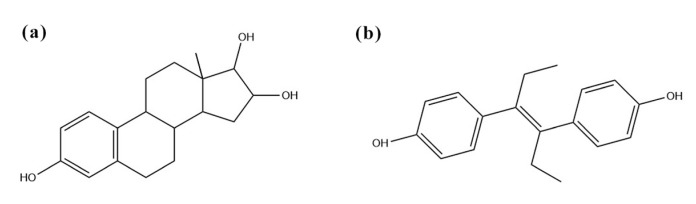
Molecular structure of E3 (**a**) and DES (**b**).

**Figure 8 ijms-23-05758-f008:**
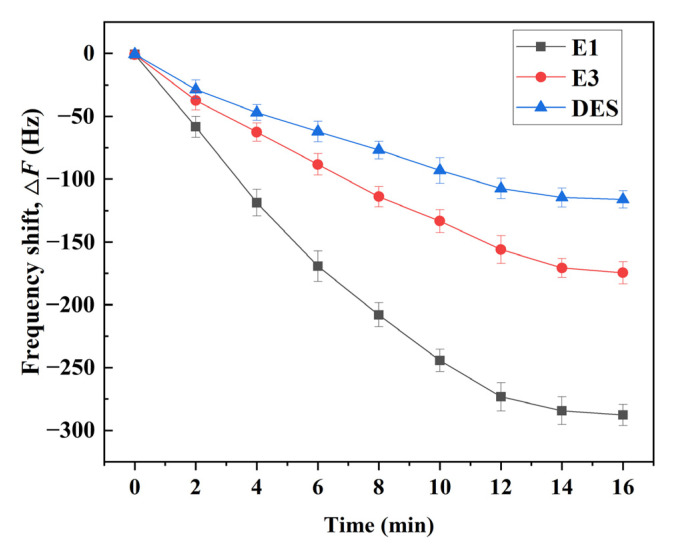
Sensor selectivity of E1, E3, and DES.

**Table 1 ijms-23-05758-t001:** The structure parameters of E1 using different methods with two basis sets 6-311+G(*d*,*p*) (I) and 6-31G(*d*,*p*) (II).

Species	B3LYP		CAM-B3LYP		LC-WPBE		M062X		PBE0		ωB97XD		Exp.
	I	II	I	II	I	II	I	II	I	II	I	II	
*R* (nm)													
C9-O19	0.1207	0.1212	0.1203	0.1207	0.1202	0.1207	0.1201	0.1206	0.1217	0.1222	0.1203	0.1208	0.1200
C8-C9	0.1539	0.1542	0.1530	0.1534	0.1525	0.1529	0.1535	0.1537	0.1541	0.1546	0.1533	0.1536	0.1541
C7-C8	0.1546	0.1547	0.1540	0.1541	0.1534	0.1536	0.1542	0.1543	0.1546	0.1547	0.1540	0.1542	0.1620
C5-C10	0.1552	0.1552	0.1543	0.1544	0.1536	0.1537	0.1544	0.1543	0.1552	0.1553	0.1545	0.1546	0.1550
C6-C7	0.1539	0.1543	0.1534	0.1536	0.1529	0.1531	0.1536	0.1536	0.1542	0.1544	0.1534	0.1536	0.1488
C12-O20	0.1371	0.1368	0.1365	0.1362	0.1361	0.1359	0.1363	0.1361	0.1375	0.1372	0.1362	0.1360	0.1388
C12-C11	0.1393	0.1397	0.1388	0.1391	0.1386	0.1390	0.1391	0.1394	0.1400	0.1404	0.1390	0.1393	0.1426
C14-C13	0.1400	0.1401	0.1395	0.1395	0.1392	0.1394	0.1397	0.1397	0.1413	0.1407	0.1396	0.1397	0.1426
*Φ* (°)													
C5-C6-C7	104.33	104.33	104.31	104.32	104.28	104.32	104.36	104.38	104.31	104.32	104.43	104.44	104.09
C7-C8-C9	105.99	105.87	105.74	105.77	105.86	105.78	105.75	105.67	106.12	105.99	105.90	105.82	101.53
C11-C12-C13	119.45	119.33	119.49	119.38	119.42	119.30	119.52	119.42	119.39	119.25	119.42	119.32	122.32
C14-C13-C12	121.32	121.36	121.26	121.30	121.19	121.28	121.20	121.23	121.38	121.42	121.27	121.29	118.93
C2-C1-C11	122.53	122.53	122.49	122.49	122.53	122.54	122.39	122.39	122.58	122.58	122.47	122.47	124.16
C5-C4-C3	110.95	110.95	110.75	110.77	110.52	110.60	110.42	110.44	110.81	110.83	110.56	110.59	109.38
C5-C6-C17	112.34	112.32	112.24	112.22	112.09	112.09	111.95	111.94	112.25	112.24	112.14	112.11	111.34

**Table 2 ijms-23-05758-t002:** Relevant parameters of the complexes formed between E1 and IA with different imprinted ratios.

Imprinted Ratios between E1 and IA	Actions Sites	Bond Length (nm)	Amount of Hydrogen Bonds	Δ*E*_1_ (kJ/mol)
1:1	O49—H50---O20	0.1838	2	−31.73
	C11—H33---O48	0.2396		
1:2	C10—H32---O48	0.2422	5	−76.24
	C4—H39---O48	0.2539		
	O49—H50---O19	0.1761		
	C11—H33---O70	0.2393		
	O71—H72---O20	0.1848		
1:3	C10—H32---O48	0.2421	7	−114.52
	C4—H39---O48	0.2551		
	O49—H50---O19	0.1802		
	C11—H33---O70	0.2355		
	O71—H72---O20	0.1858		
	C8—H21---O78	0.2319		
	O79—H80---O19	0.1796		
1:4	C10—H32---O48	0.2455	8	−141.55
	C4—H39---O48	0.2552		
	O49—H50---O19	0.1801		
	C8—H21---O78	0.2312		
	O79—H80---O19	0.1792		
	O20—H42---O63	0.1927		
	O64—H65---O20	0.1880		
	C7—H28---O93	0.2563		

**Table 3 ijms-23-05758-t003:** Δ*E*_3_ values of different solvents and the corresponding hydrogen bond length.

Solvent	Hydrogen Bond Length (nm)	Δ*E*_3_ (kJ/mol)
	C11—H33---O48	
H_2_O	0.2447	−59.37
ACN	0.2447	−57.82
MT	0.2447	−54.92
THF	0.2435	−48.02
TL	0.2412	−28.33

**Table 4 ijms-23-05758-t004:** Solubility of the reaction components and adsorption capacity of E1-MIPs in different solvents.

Solvent	Solubility of Reaction Components	Adsorption Capacity of MIPs (mg/L)
H_2_O	Insoluble	—
ACN	Easily soluble	2.02
MT	Easily soluble	2.58
THF	Slightly soluble	—
TL	Insoluble	—

**Table 5 ijms-23-05758-t005:** The hydrogen bond lengths, *ρ*(r)_bcp_, ▽^2^*ρ*(r)_bcp_, and E_H_ values of the complex formed between E1 and IA.

Molar Ratios	Actions Sites	Hydrogen Bond Length (nm)	*ρ*(r)_bcp_ (a.u.)	▽^2^*ρ*(r)_bcp_ (a.u.)	*E*_H_ (kJ/mol)
1:1	O49—H50---O20	0.1838	0.0304	0.1229	−35.58
	C11—H33---O48	0.2396	0.0105	0.0353	−8.27

## Data Availability

Not applicable.

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
