# Peer review of "Study on the Preparation of Estrone Molecularly Imprinted Polymers and Their Application in a Quartz Crystal Microbalance Sensor via a Computer-Assisted Design"

_ijms, 2022, doi:10.3390/ijms23105758_

Round 1

Reviewer 1 Report

In this paper, authors investigated the natural bonding orbital charges and molecular electrostatic potential of estrone E1 and itaconic acid with the help of DFT to analyze the interaction active sites and screen the optimal interaction ratio, cross-linker, and solvent of the E1-IA imprinting system. Due to E1 is an important environmental estrogen that interferes with the reproductive system of aquatic organisms even at extremely low concentrations in environmental water, the development of reliable and sensitive systems for its detection is of interest, and here E1-MIPs modified QCM sensor had been constructed to detect E1.

Comments regarding the content:

  • In general, I think the article is well written, although there is an absolute abuse of a multitude of acronyms for molecules, particles and processes that should be organized in an annex or appendix with a list of abbreviations. If we also take into account that IJMS is a journal for a general audience, this is even more important.
  • Most of the parameters present 4 decimals. Is the calculation thereof so precise as to give so many decimals? The number of decimals that are put in the data must be justified.
  • No value of the estimated with calculated error appears, why?
  • You give some values of E1-MIPs and NIPs sizes and say in the text “E1-MIPs and NIPs were homogeneous microspheres and uniform distribution”, but you do not show representative SEM images or the data that you use to estimate these data, and affirm they are homogeneous. You should show at least 2 images showing several particles and one particle amplified in an inset for both samples; and also the distributions and fittings you have used to calculate size data. Regarding methods, you should also describe how the samples were prepared and measured in SEM measurements.
  • I also wonder if E1-MIPs and NIPs present aggregation. Did you see this in SEM or DLS measurements?
  • The results seem to confirm the fabrication of a sensor to target E1 with a certain sensibility, but I miss the comparison of the parameters obtained with this sensor and the ones obtained with other current methods to see the possible advantages.

Comments regarding style:

  • In general, figures should be bigger to improve quality, in particular in Figure 6, the numbers and characters can´t be read.
  • Caption of Figure 4, E1-CA appears as a not readable legend.

Reviewer 2 Report

I think this paper presents scientific data and data analyis. There are just some minor issues.

Figure 4, the y axis label is delta E while the caption used delta E2. The Chinese words in the legend may need to be corrected.

Figure 6, how was the distribution data acquired?

Reviewer 3 Report

Section 2.4.: Can the statement saying “To achieve a better imprinting effect, the interaction between the selected cross-linker and IA functional monomer should be greater than that between the cross-linker and E1 when preparing MIPs.” have a reference backing up this information?

The legend in figure 4. Needs fixing.

During the modelling of the solubility of the different components: Surely temperature has an effect on the solubility, was this factored into the calculations? If so, what temperature were the calculations ran at? If not, can this be implemented to make the calculations more reflective of reality?

From the solubility study you state “Therefore, MT was chosen as the solvent for the preparation of E1-MIPs in this experimental study.” And then when it comes to the synthesis of the MIPs you state “E1-MIPs (NIPs) were synthesized at 333 K, when PETA was as cross-linker, toluene was as solvent”. Why is toluene now selected as the solvent?

During the selectivity study the sensor was left to stabilize over 16 minutes at one concentration. It would be preferable to see the performance of the sensor over a concentration range of each structural analogue instead.

Regarding the last comment: Was the same stabilization period also used for the QCM measurements with E1? If so, this should be mentioned in the experimental section.  

Round 2

Reviewer 1 Report

I think the revised paper version has improved the general quality.

A point needs to be made; the authors should locate preparation of samples and analysis procedure for SEM measurements in the Methods section, in “3.4. Preparation of E1-MIPs and NIPs. Characterization by Scanning Electron Microscopy” or in a new “3.5. Characterization of E1-MIPs and NIPs by Scanning Electron Microscopy”; and the corresponding results can be in the current section 2.7 in Results.

Author Response

  1. A point needs to be made; the authors should locate preparation of samples and analysis procedure for SEM measurements in the Methods section, in “3.4. Preparation of E1-MIPs and NIPs. Characterization by Scanning Electron Microscopy” or in a new “3.5. Characterization of E1-MIPs and NIPs by Scanning Electron Microscopy”; and the corresponding results can be in the current section 2.7 in Results.

Response: We are very agreed with the Referee’s suggestion. According to the Referee’s suggestion, the description had been added in Lines 390-395 of manuscript.

3.5. Characterization of E1-MIPs and NIPs by scanning electron microscopy 

    The E1-MIPs (NIPs) was dispersed in methanol, and the mixed solution was put in ultrasonic for 40 min to make the polymer evenly dispersed in methanol. Then the mixed solution was dropwise dripped onto the silicon wafer and let the silicon wafer dry naturally. After gold spray, the morphology and particle size of E1-MIPs (NIPs) were observed by SEM.

Reviewer 3 Report

The corrections are of sufficient quality. 

Author Response

1. The corrections are of sufficient quality.
Response: Thank you very much.